# Raman Spectroscopy of Carotenoid Compounds for Clinical Applications—A Review

**DOI:** 10.3390/molecules27249017

**Published:** 2022-12-17

**Authors:** Joy Udensi, James Loughman, Ekaterina Loskutova, Hugh J. Byrne

**Affiliations:** 1FOCAS Research Institute, Technological University Dublin, City Campus, Camden Row, Dublin 8, D08 CKP1 Dublin, Ireland; 2School of Physics and Clinical and Optometric Sciences, Technological University Dublin, City Campus, Grangegorman, Dublin 7, D07 EWV4 Dublin, Ireland; 3Centre for Eye Research, Ireland, Technological University Dublin, City Campus, Grangegorman, Dublin 7, D07 EWV4 Dublin, Ireland

**Keywords:** carotenoids, beta carotene, lycopene, lutein, zeaxanthin, Raman spectroscopy, skin, optometry, blood

## Abstract

Carotenoid compounds are ubiquitous in nature, providing the characteristic colouring of many algae, bacteria, fruits and vegetables. They are a critical component of the human diet and play a key role in human nutrition, health and disease. Therefore, the clinical importance of qualitative and quantitative carotene content analysis is increasingly recognised. In this review, the structural and optical properties of carotenoid compounds are reviewed, differentiating between those of carotenes and xanthophylls. The strong non-resonant and resonant Raman spectroscopic signatures of carotenoids are described, and advances in the use of Raman spectroscopy to identify carotenoids in biological environments are reviewed. Focus is drawn to applications in nutritional analysis, optometry and serology, based on in vitro and ex vivo measurements in skin, retina and blood, and progress towards establishing the technique in a clinical environment, as well as challenges and future perspectives, are explored.

## 1. Introduction

Carotenoids encompass a wide range of fat-soluble pigmented compounds (Figure 1) and are the most widely occurring pigments in nature [1]. They are present across various parts of the ecosystem, including in plants, animals and even micro-organisms. Dietary sources in humans include carotenoid-rich fruits, vegetables, animal fat and other foods. Of the 42 dietary carotenoids, 14 are absorbed, circulated in the blood and deposited in tissue [2], where they play important roles in biological function, particularly as antioxidants and photoprotective agents [3]. Carotenoid consumption is associated with reduced risk of some cancers, including breast, oesophageal and lung [4,5,6,7], coronary heart disease [8,9], stroke [10,11], type 2 diabetes mellitus [12,13,14,15] and asthma in adults and children [16].

Beta carotene is the most common form of carotenoid in plants and, along with lycopene, is the most abundant in human blood (Table 1) [17]. It is one of the primary sources of vitamin A (retinol) and thus plays a critical role in biological function at the level of cells and organs [18]. Excess beta carotene is stored in fatty tissue, including skin, and can lead to carotenoderma [19]. Lutein (L), zeaxanthin (Z) and meso-zeaxanthin (MZ) (Figure 1) accumulate in several tissues of the human body but are found in the highest concentration in the eye (where they accumulate at the macula to form macular pigment (MP)) and in the brain [20]. These carotenoids are believed to exert a favourable effect on neuronal processing in the eye and in the brain and to confer neuroprotective benefits against ocular diseases such as age-related macular degeneration (AMD) [21,22,23], glaucoma and diabetic retinopathy [24,25,26]. Carotenoids play a major role in combating oxidative stress, a major factor in AMD [27] and glaucoma [28], and in doing so, are degraded. As dietary compounds, supplementation with lutein, zeaxanthin and meso-zeaxanthin has been demonstrated to successfully augment the MP in glaucomatous eyes [29]. Monitoring carotenoids and associated oxidative stress markers in humans is therefore of great importance, and when it is performed (especially in clinical studies), blood plasma or serum is used. Skin carotenoid measurements are also becoming increasingly popular for monitoring fruit and vegetable intake, especially in children [30,31], to ascertain overall nutritional health. Although they vary with dietary intake, mean concentrations of the most abundant serum and skin carotenoids have been documented by a systematic review of published scientific literature and are represented in Table 1 [17] and Table 2 [32], respectively.

Reduced overall carotenoid levels can be associated with disease, potentially due to oxidative stress. These overall estimates can be generally used for medical diagnostics [34,35,36] without necessarily knowing the details of the type of carotenoid present or their absolute or relative quantities. However, quantitative analysis of the dietary carotenoids can become of great relevance to areas such as nutritional assessment, whereby the dietary carotenoid content of the body is linked to general nutritional status [18]. Moreover, the identification and quantification of specific carotenoids can advance the understanding and appreciation of the individual components and their specific roles, such as the possible contribution of macular carotenoids to ocular health, nutrition and disease prevention and treatment.

The gold standard technique for analysis of carotenoid content in the blood is high-performance liquid-chromatography (HPLC), which is costly and laborious, and, while it provides a quantitative analysis, does not easily differentiate structurally similar carotenoids such as lutein/zeaxanthin [37,38,39,40]. A number of alternatives involving colourimetric and spectrometric techniques have more recently emerged [37,41,42]. As conjugated highly polarisable organic molecules, carotenoids are particularly active in Raman spectroscopy and, although only present in relatively low concentrations, can dominate the spectrum of blood serum, particularly using visible lasers as a source, at whose wavelengths they are resonantly or near resonantly enhanced [43,44]. Raman spectroscopy has become popular for monitoring dermal carotenoid content [30,31,36] and has been used to correlate dermal and blood carotenoid content [45,46].

After reviewing the structure and optical properties of carotenoids in Section 2, this paper examines the literature describing the work to date, exploring potential clinical applications based on Raman spectroscopy of carotenoid compounds, focussing particularly on beta carotene, lutein and zeaxanthin, and applications involving their differentiation and quantification. It is based on a Web of Science abstract search (30 August 2022) of the terms (Raman) AND (Carotene OR Lutein OR Zeaxanthin), excluding the terms (plant OR oil). Relevant studies were identified from amongst the 453 returned articles and are discussed under the headings of serum/plasma, skin and eyes/macula.

## 2. Structure and Properties

### 2.1. Structure

Carotenoids are hydrocarbons made up of a basic polyene backbone structure, a hydrocarbon chain of 40 sp^2^ hybridised carbons, the structure of which is represented as alternating double and single bonds between the carbon atoms (Figure 1). The π electrons are highly conjugated, giving rise to relatively broad molecular orbitals (MO) and low energy transitions in the visible region that produce their characteristically strong colouring [47,48]. Most naturally existing carotenoids have a trans configuration throughout their conjugated double bonds [49].

Carotenoids are broadly divided into two groups, carotenes, which contain only carbon and hydrogen atoms, e.g., alpha and beta carotenes, and a second group known as the xanthophylls; they contain oxygen atoms in addition to hydrogen and carbon atoms in their structures, e.g., lutein and zeaxanthin [50]. Both forms are poorly soluble in aqueous media and tend to aggregate in J (head to tail) or H (stacked) aggregates [51]. Xanthophylls contain at least one hydroxyl group and are, in general, more polar than carotenes. This has implications for how they aggregate [52,53,54,55] and are transported. In blood circulation, beta carotene and lycopene tend to be predominately localised in the low-density lipoproteins, while lutein and zeaxanthin are more evenly distributed among both low and high-density lipoproteins [56,57].

### 2.2. Optical Properties of Carotenoids

The absorption spectra of carotenoids are dominated by strong π−π* transition in the visible region of the spectrum, the wavelength positioning of which increases with increasing conjugation length [47,58]. For beta carotene in pure ethanol, the absorption maxima at 483 nm, 453 nm and 427 nm (Figure 2, Table 3) are, respectively, attributed to the 0–0, 0–1 and 0–2 transitions of the S_0_ (1^1^Ag)-S_2_ (1^1^Bu) vibrational manifold, transitions to the first S_1_ (2^1^Ag) state being forbidden due to symmetry restrictions [48,54]. S_2_ state excitation rapidly decays by internal conversion to the S_1_ state, radiative relaxation from which is similarly symmetry forbidden, and thus carotenoids exhibit negligible fluorescence emission [59].

Table 3 highlights some important optical and chemical properties of dietary carotenoids, including their absorbance maximum, main Raman peak positions, conjugation length, number of hydroxyl groups and conjugated double bonds. Since the position of the longest wavelength absorption transition of carotenoids should normally be directly proportional to their conjugation length [58], zeaxanthin, which is the longer xanthophyll, should be shifted to the red when compared with lutein, especially if all other conditions, such as solvent type and temperature are kept constant for both carotenoids [58,60]. However, because of their lipophilic nature, carotenoids tend to aggregate in hydrophilic environments via weak intermolecular forces such as van der Waals interactions, hydrogen bonding and dipole forces [61]. This results in shifts in the absorption spectrum towards the red or blue region [51], depending on whether they form strongly (H) or weakly (J) coupled aggregates [61,62].

**Table 3 molecules-27-09017-t003:** Summary of some optical and chemical properties of carotenoids.

Carotenoid	Absorbance λ Max in Ethanol (nm) [55,63,64,65]	Main Raman Peaks Positions (cm^−1^) [66]	Conjugation Length (*n*) [60,67]	Hydroxyl Groups	Number of Conjugated Double Bonds [68,69,70]
Beta carotene	~427, 453, 483	~1000, 1160, 1520	9.6	none	11
Lycopene	~447, 474, 504	~1000, 1160, 1520	11	none	11
Lutein	~424, 445, 472	~1000, 1160, 1520	9.3	2	10
Zeaxanthin	~424, 445, 472	~1000, 1160, 1520	9.6	2	11

The longest wavelength absorption maximum of beta carotene is shifted from ~476 nm in ethanol to ~515 nm in 1:1 ethanol–water solution due to J aggregation, while the 453 nm and 427 nm peaks are blue-shifted, attributed to H-aggregation [52,53,54,55]. Similar spectral shifting in a range of solvents has been documented depending on polarising efficiencies [71,72,73]. The UV-visible absorption spectrum of beta carotene in solution is also pressure dependent, with the longest wavelength absorption maximum shifting as far as ~580 nm in carbon disulphide solution at 0.96 GPa [74]. In H aggregates of zeaxanthin, prepared in ethanol:water mixtures, the UV-vis spectrum is dominated by a strong feature at 370 nm, while in the J aggregate, formed in 1:9 THF:water solutions, the longest wavelength absorption maximum is shifted from 485 nm to ~510 nm [75,76]. A similar red shift is seen in J aggregates of lutein [77].

Both beta carotene and the xanthophylls, lutein and zeaxanthin, when dissolved in hydrated organic solvents, can form either of the two kinds of aggregates depending on the nature of the solvent [62]. For instance, when incorporated into lipid bilayers, xanthophylls appear mostly in the monomeric form at concentrations below 0.5 mol%. With higher concentrations, they tend to form mostly H aggregates [62]. It was also recently shown that as a result of carotenoid–protein interactions, absorption changes consistent with J-type aggregation can occur [51,62]. The presence of two hydroxyl groups in a carotenoid (e.g., lutein and zeaxanthin) generally promotes the formation of the H-type of aggregates [51]. When the hydrogen-bond formation is intercepted, for example, by esterification or a lack of end-ring functional groups, J-type aggregates could be formed [51]. Additionally, zeaxanthin is less polar than lutein, making it easier to form H-type aggregates in water/ethanol mixtures than lutein [51,62].

In the bloodstream, carotenoids are predominantly associated with lipoproteins, which are bound by albumin [43,78,79]. In addition to the solubilising effect, facilitating transport around the body, the complexation has been shown to protect the electron-rich molecules from oxidation [80]. When complexed with bovine serum albumin (BSA), the longest wavelength absorption maximum of beta carotene has been observed at ~515 nm, indicating the presence of J aggregates [80,81]. A similar absorption profile was observed for zeaxanthin, while lutein showed a strong, blue-shifted absorption maximum, characteristic of H-aggregates [81]. In beta carotene-bound albumin nanoparticles, the absorption profile was seen to extend further to the red, towards 600 nm [82].

## 3. Raman Spectroscopy of Carotenoids

Raman spectroscopy is based on the inelastic scattering of light (photons) by (quantised) vibrations of a molecule, such that the oscillating electric field of the light interacts with the oscillating electric polarisation of the bond, resulting in a transfer of a quantum of vibrational energy to or from the photon to the molecule [83]. The phenomenon depends on the real component of the material polarisability (refractive index) and can occur in the absence of molecular absorption (non-resonant), in which case the scattering process is often represented as a transition to a virtual state and instantaneous return to a higher vibrational state, inelastically scattering the photon of energy which has been reduced by the vibrational energy difference (Stokes Raman scattering) (Figure 3) [84,85]. This energy difference is equivalent to that typically probed by infrared (IR) spectroscopy, but because of the differing physical phenomena of IR absorption, an electric dipole transition, and Raman scattering, dependent on molecular polarisability, the techniques are considered to be mutually complementary [83]. When the Raman source is at or close to electronic absorption, the real component of the polarisability is enhanced according to the Kramer–Kronig relations [86], giving rise to the phenomenon commonly called resonance Raman spectroscopy (RRS), by which some Raman features of a spectrum can be enhanced by orders of magnitude [82,84,85,86]. However, note that the process is still one of light scattering rather than an absorption and re-emission process, as is the case for fluorescence, for example. Whether a photon is absorbed or scattered (elastically or inelastically) depends on the relative probabilities (cross-sections) of the processes in the material at the wavelength in question. Therefore, in order to understand how best to differentiate and quantify carotenoids using Raman spectroscopy, it is important to consider the best laser energy source wavelength to use to provide the best outcomes.

Raman spectra of π conjugated organic molecules are dominated by the modes coupled to the highly polarisable π electron system, even more so when the source wavelength is near or at the electronic resonance, which can give rise to several orders of magnitude increase in the response. For natural carotenoids, independent of the end group, the Raman spectra are dominated by three characteristic bands at ~1520 cm^−1^ (*v*_1_), ~1160 cm^−1^ (*v*_2_) and ~1000 cm^−1^ (*v*_3_) (Table 3, Figure 4) [66,87]. The *v*_1_ line derives from the conjugated C=C stretching vibrations, and its frequency positioning can be used to determine the degree of conjugation of the π-electron system. The *v*_2_ line, located at ~1160 cm^−1^, has been widely assigned to a combination of C-C stretching and C-H in-plane bending vibrations. The third most prominent, *v*_3_, the line at ~1000 cm^−1^, is associated with CH_3_, in-plane rocking vibrations [84,88].

The dominance of these three bands in the Raman spectrum of carotenoids, particularly when the source laser wavelength is at or near resonance, gives rise to a characteristic and recognisable carotenoid signature, evident, for example, in the 532 nm Raman spectrum of foodstuffs [35], blood serum [43,44] and algae [89], which has attracted interest for planetary exploration [90]. As the 9 C=C polyene backbone is common to many carotenoid compounds of interest, it is difficult to differentiate them [60]. However, the region between 1000 and 1400 cm^−1^ also contains a series of weaker features which are not resonantly enhanced, mainly assigned to C-H in-plane bending modes. Below 1000 cm^−1^, there is an additional feature at ~950 cm^−1^ called the *v*_4_ band, which arises from out-of-plane motions of the H nuclei on the conjugated chain. Although relatively weak, it can increase in intensity if there are distortions in the plane modes of the carotenoids, for example, due to steric hindrance within a protein binding pocket [91]. A few other weaker features, resonantly enhanced in the 532 nm spectra, also contribute to the Raman spectra and can be found in the region > 2000 cm^−1^. These weaker features, at ~2169 cm^−1^, ~2526 cm^−1^, ~2679 cm^−1^ and ~2920 cm^−1^, are overtone and combination bands which arise from the total symmetric character of the carotenoid vibrations (*v*_1_, *v*_2_ and *v*_3_) [92,93].

Figure 4a also shows the features of beta carotene under different excitation conditions. For all wavelengths, the spectra clearly exhibit the characteristic carotenoid peaks at ~1519 cm^−1^, ~1158 cm^−1^ and ~1004 cm^−1^. The blue (473 nm) and green (532 nm) regions show high absorbance, the green being near resonance; thus, the Raman scattering process is expected to be resonantly enhanced. The red (660 nm) and near-infrared (785 nm) show much lower absorbance and are thus near- and/or off- resonance, respectively [48,84].

The weaker features, such as those at ~2126 cm^−1^, ~2679 cm^−1^ and ~3036 cm^−1^, appear to be extinct or very poorly visible in the 785 nm, 473 nm and 660 nm spectra but are all quite prominent in the 532 nm spectrum. The near-resonant/resonant excitation wavelength can therefore be said to be the best choice for accurate and optimum qualitative and quantitative analytical deductions of beta carotene and other carotenoids. Despite this, the relative strengths of the carotenoid features will depend on the resonance conditions of the source wavelength relative to the material absorption [48,84].

## 4. Differentiation and Quantification of Carotenoids

The identification and quantification of dietary carotenoids using Raman spectroscopy can advance studies in healthcare and nutrition. For instance, differentiating lutein and zeaxanthin can become very important in macular pigment clinical research, especially if this difference can be directly detected and subsequently quantified in the living human retina. A big step towards achieving this would be to correlate the differences between model laboratory experiments and the actual carotenoid contents of human tissues. In general, simple and precise artificial samples are designed to establish and verify spectroscopic methods for carrying out in vivo studies, such as the analysis of carotenoids in living tissues. For example, the standard reference sample for the analysis of the carotenoid content of human skin is usually a solution of beta carotene in an organic solvent such as ethanol or acetone [55,61]. However, a solution of a carotenoid does not correctly simulate the characteristics of carotenoids in human tissues. Carotenoids are mostly found in the lipid membranes of the cells and tissues in the body, which mainly consist of water [55]. Physiological alternatives such as emulsions, hydrogels or aqueous solutions of serum albumin may be considered more realistic [55]. The proper preparation of samples for the identification and quantification of carotenoids in human tissue should therefore be based on how well the optical properties of the actual tissues and reference samples can be replicated. However, preparing carotenoid solutions to resemble their physiological state can be quite challenging. Firstly, carotenoid powders are insoluble in water, and the methods for preparing them in aqueous solution and hydrogels can affect their spectral properties, because of problems with the stability of the solution, formation of aggregates and degradation by oxidation due to the aqueous environment [55]. Dietary carotenoids are also highly similar in structure and differentiating them using Raman spectroscopy can be somewhat challenging, as they have almost identical signatures [60].

Ideally, the carotenoids can be separated based on their conjugation length and absorption properties. Structurally, xanthophylls (lutein and zeaxanthin) differ from carotenes (e.g., beta carotene) mainly because of the presence or absence of oxygen atoms in their chains (Figure 1) [84]. Therefore, it is expected that lutein and zeaxanthin should be the most difficult to distinguish, as they differ only in the length of their conjugated chain. This is seen in one of their cyclic end groups, in which there is a difference in the position of a double bond for lutein, causing it not to be conjugated in that position [60,84]. Beta carotene, on the other hand, can also be difficult to distinguish. It is the most abundant carotenoid in the human body and has similar electronic and spectroscopic characteristics to the macular carotenoids, lutein and zeaxanthin. They all have cyclic end groups containing double bonds, which are expected to be conjugated with the nine double bonds of the linear chain. In fact, beta carotene is similar to zeaxanthin in that they both contain two C=C conjugated bonds in their cyclic end groups [58]. Therefore, it is expected that their conjugated chain lengths would be the same, again as derived from the frequency of the *v*_1_ band. From the literature, the estimated conjugated length of beta carotene and zeaxanthin is *n* = 9.6, while it is 9.3 for lutein (Table 3), differing by 0.3 since ring double bonds add approximately 0.3 effective C=C to the entire chain length [58,60]. Distinguishing beta carotene from xanthophylls (especially zeaxanthin) using Raman spectroscopy relies on identifying and analysing small shifts in the spectral features, especially the *v*_1_ band located at ~1520 cm^−1^, which derives from the carbon-to-carbon double bond stretching modes (C=C) of carotenoids and directly correlates with their conjugation lengths [60]. For example, Arteni et al. [60] evaluated the carotenoid status of the macular pigment from retinal samples ex vivo with the aim of distinguishing the carotenoids lutein and zeaxanthin. The study compared the resonance Raman spectra of the retina samples with those of lutein and zeaxanthin solubilised in tetrahydrofuran (THF), taken at different excitation wavelengths (488.0, 501.7, 514.5 and 528.7 nm). Both lutein and zeaxanthin were found to contribute to the spectra at all wavelengths explored. Notable shifts were seen in the *v*_1_ (1525 to 1521 cm^−1^) position as the excitation moved from 488 nm to a higher excitation of 514.5 nm. This shifting was evidence that the macular carotenoids each dominated at a different wavelength and when compared to spectra from in vitro carotenoids, lutein was determined to be dominant at 488 nm while zeaxanthin entered into resonance at a higher excitation of 514 nm [60]. Other supporting studies in which lutein and zeaxanthin complexed with binding proteins were analysed also showed an absorption maximum at 482 nm for lutein [94] and at 510 nm for zeaxanthin [95]. In combination, these studies indicate that, taking advantage of the appropriate excitation wavelengths and absorption properties, xanthophylls can be distinguished.

However, as described in Section 2.2, carotenoids are prone to aggregation in complex and particularly biological environments, and the position and strengths of the major carotenoid Raman bands can also be affected by the shifts in absorption [51,60]. For instance, in the study of Arteni et al. [60], when the spectra of isolated carotenoids were compared to those of the retinal tissue, the shift in the *v*_1_ position for the different excitation wavelengths employed was seen to be greater in the retinal tissue [60]. Therefore, in the attempt to identify, analyse and quantify the carotenoid content of biological samples, it is important to use model samples which, as closely as possible, mimic the biological environment of interest for comparison.

It should be noted that a lot of literature reports which identify carotenoid Raman peaks identify the carotenoid as beta carotene [17,96,97,98,99,100,101], often based on other literature assignments [102]. However, without performing accurate identification/differentiation analysis, the carotenoids observed are more likely a combination of beta carotene and other carotenoids. To ensure the correct identity of the compounds in solution, relevant factors such as solvent type, solvent/water ratio, carotenoid concentration, temperature and pH need to be standardised and monitored carefully throughout a study. The appropriate application of multivariate techniques, such as principal components analysis (PCA) and linear discriminant analysis (LDA), is used to analyse multi-dimensional data sets to reduce the number of variables to enhance the accuracy of identification [103].

In addition, to fully understand the physiological importance of carotenoid content, there is a need to quantify them accurately. Clinicians mostly rely on serum HPLC for quantification, despite the complexities of the technique. Raman spectroscopy can be used reliably to quantify carotenoids by accurately measuring the carotenoid Raman signals from the identified dietary carotenoids. This signal can often be affected by self-absorption, a situation where the wavelength of the Raman source is resonant with the sample absorption, leading to a reduction in the intensity as it propagates through the sample, and the Raman scattered light itself is similarly attenuated by the sample absorption [48,104]. This self-absorption process can result in a reduced Raman signal and false measurements, especially when the analyte is resonantly enhanced, as is common with carotenoids [48,104]. Hence, Raman measurements must be adjusted for self-absorption during quantification.

Despite the challenges, there have recently been some laudable advancements in quantifying dietary carotenoids, especially skin carotenoids, beta carotene and lycopene [30,105]; however, the prospects for the utilisation of the technique in clinics are still somewhat distant. This review further discusses advances that have been made in the application of Raman spectroscopy of carotenoids to meet clinical needs, paying particular attention to the importance of quantification and differentiation and how they can further positively transform the diagnosis and treatment of various diseases.

## 5. Raman Spectroscopy of Carotenoids in Biological Systems

### 5.1. General Biological Systems

Dietary Carotenoids exist mostly in the skin, eyes and blood, and Raman spectroscopy of these systems will be discussed in detail. Despite this, there are other parts of the body for which the application of Raman spectroscopy has been explored in depth. For instance, carotenoids are found in human bone and surrounding fatty tissue, both in significant and individually variable concentrations [106,107]. Measurements of biopsied tissue samples with molecule-specific Raman spectroscopy and HPLC reveal that all carotenoids that are known to exist in human skin, including lycopene, β-cryptoxanthin, lutein and zeaxanthin, are also present in human bone [107]. An RRS study carried out in 2013 on bone tissue, using an argon laser as an excitation source, showed that all three major Raman peaks, characteristic of carotenoids, are visible in the spectrum of bone [107].

The interaction between hydroxyapatite (HA), a major component of bone, and carotenoid has been demonstrated in a time-lapse Raman imaging study of the mouse osteoblastic mineralisation process at subcellular resolution [108]. KUSA-A1 cells were stimulated to differentiate into osteoblasts, and this was followed by 4 hourly Raman imaging for a duration of 24 h. The Raman images acquired showed the action of carotenoid from bands at 960 cm^−1^, 1008 cm^−1^, 1160 cm^−1^ and 1526 cm^−1^, which were visible at 0 h and diminished as the time lapsed [108].

In the study and treatment of cardiovascular diseases, there is now ample evidence that carotenoids have beneficial effects on some of the debilitating factors implicated in the development of the disease, including oxidative stress, inflammation, dyslipidaemia and thrombosis [18]. Raman spectroscopy has been applied in a number of studies in the quest for a more effective and accurate technique for diagnosing the disease. Some of these studies include the spectral diagnosis and analysis of a superior vesical artery calcification, whereby calcification deposits in the superior vesical artery were qualitatively confirmed to contain beta carotene as a major component by Fourier transform infrared and Raman micro spectroscopies [109]. Raman analysis revealed two strong vibrational bands at 1516 and 1158 cm^−1^ from carotenoids, thus presenting evidence that could be used to diagnose this condition [109].

Another study examined the necrotic core regions of atheromatous plaque, and the yellow crystals seen were revealed by Raman microspectroscopy to be rich in carotenoids [110]. The technique was applied to analyse individual cellular and extracellular components of atherosclerotic lesions in different stages of disease progression in situ, using a high-resolution confocal Raman spectrometer with 830 nm laser light. The Raman spectra revealed two prominent carotenoid bands in sections of the morphological structure of the coronary artery, including collagen fibres, foam cells, smooth muscle cells, etc. [110]. A similar study of atherosclerotic lesions combined fluorescence lifetime imaging (FLIm) with Raman spectroscopy [111]. In this approach, six human coronary artery samples were imaged using a FLIm and Raman multimodal fibre optic probe at 785 nm excitation. Raman data were analysed using an end-member technique and compared with histological findings. With the help of a descriptive modelling approach based on multivariate analysis, two carotenoid bands were identified at 1155 and 1521 cm^−1^ and their intensity was seen to increase with increases in LDL protein in atherosclerotic regions of the heart [111]. The combination of Raman spectroscopy with reflectance and fluorescence to provide detailed biochemical information in tissues has also been shown to detect atherosclerotic plaque features with high sensitivity and specificity, revealing the characteristic Raman properties (all three peaks) of carotenoids [112]. There is also Raman spectroscopic evidence revealing the interplay between atherosclerotic and medial calcification in the human aorta. In the study by You and colleagues [113], a reduction of carotenoid was discovered in the whole aortic intima of atherosclerotic tissues of volunteers as compared with non-atherosclerotic tissues when excited with a 785 nm laser source. Carotenoid was revealed by its strong Raman bands at 1157 and 1526 cm^−1^, adding to the store of evidence describing the potentially protective effects of carotenoids against atherosclerotic plaques [113].

Another area of interest is the application of Raman spectroscopy of carotenoids to investigate cancerous tissues. In the study of oral tumours, for instance, the efficacy of Raman spectroscopy to differentiate anatomical sites was explored. Eighty samples, including tumour and normal tissues, were cryopreserved from three different sub-sites: the tongue, the buccal mucosa and the gingiva of the oral mucosa during surgery. Raman imaging at 532 nm showed a prominent carotenoid peak at 1518~1524 cm^−1^, evident in both normal and tumour samples but with much lower intensity in normal samples. Linear discriminant analysis (LDA) and quadratic discriminant analysis (QDA) were used with principal component analysis (PCA) to classify the samples with statistically relevant accuracy, and the model studies revealed beta carotene variations as one of the major biomolecular difference markers for detecting oral cancer [114].

Breast cancer studies have also explored the potential of Raman spectroscopy. An example is the use of confocal Raman microspectroscopy to image unstained tissue sections of more than 60 samples of normal, benign and malignant breast tissues using an 830 nm laser source. The resultant Raman images from this study revealed the presence of only one prominent carotenoid peak at 1540 cm^−1^ [115]. Over time, the use of Raman spectroscopy in early breast cancer diagnosis has advanced even further, with more sophisticated methods coming to light. In 2020, Zheng et al. used a confocal Raman system with a 633 nm laser to examine precancerous breast lesions of volunteers with atypical hyperplasia, a very early stage of breast cancer progression. Using shell-isolated nanoparticles (SHINs) to enhance Raman signals, the study revealed the presence of slightly shifted carotenoid peaks at 1156 cm^−1^ in these precancerous lesions [116]. This is added evidence that Raman spectroscopy can detect characteristic molecular changes in breast lesions such as carotenoid intensities and may be useful for the non-invasive early diagnosis and for investigating the mechanism of breast tumorigenesis.

The application of Raman spectroscopy of carotenoids has also become relevant in the dentistry field. An increase in carotenoid levels has been revealed in patients affected by periodontal inflammation [117]. When the biopsy of periodontal ligament samples of premolars extracted for orthodontic reasons and the gingival crevicular fluid (GCF) samples were imaged by micro-Raman spectroscopy using a 633 nm laser source, elevated carotenoid levels were revealed in patients affected by periodontal inflammation as compared to normal patients. This was based on the C=C carotenoid peak at 1537 cm^−1^ due to the formation of isomerisation products containing groups related to an increase in degraded carotene in GCF, a fluid that contributes to the defence system against bacteria and their metabolites in the periodontal space [117].

In the studies described above, the authors explored the potential of Raman spectroscopy in more of a general approach and purely for diagnostic purposes, i.e., differentiating disease conditions from normal. The disease was mostly characterised by a reduction in the intensity of carotenoid Raman peaks, showing the correlation between dietary carotenoid levels and various diseases. The relevance of quantification and identification of carotenoids becomes more pronounced in biological systems in which their function is better defined, such as the skin, eyes and blood.

### 5.2. Skin

The human skin has a balanced and effective defensive mechanism, within which carotenoids play a significant role [118]. They can be found in the epidermis and dermis and are deposited by infusion from the adipose tissue, blood and lymph flows, and the secretion via sweat glands and/or sebaceous glands onto the skin surface and their subsequent penetration [119]. The predominant long-chain carotenoids in the skin are lycopene and beta carotene, accounting for 60–70% of total carotenoid content [120]. They act as antioxidants and scavengers for free radicals, singlet oxygen and harmful reactive oxygen species, which can cause skin malignancies and other diseases [118]. Carotenoids in the skin can also be important biomarkers for nutritional status [30,31] and excessive smoking [30]. They are also implicated in diseases such as erythropoietic protoporphyria, a photosensitive disorder, and can be supplemental for aiding the delay of erythema [121].

Carotenoids have been shown to modulate the skin’s response to ultraviolet radiation and contribute to defence against some of the deleterious effects of solar radiation and environmental hazards [118,122]. Skin antioxidant measurements provide a chance for early intervention strategies such as increasing dietary intake of fruits and vegetables [123], smoking cessation [30] and prescription of dietary antioxidant supplements [124,125].

#### 5.2.1. Raman Instrumentation and Quantification of Skin Carotenoids

Although HPLC, the gold standard technique for measuring carotenoids, is suitable for serum, it is considerably less so for skin because biopsies of relatively large tissue volumes are required [126]. Additionally, serum antioxidant measurements are more suited for short-term dietary intakes of antioxidants rather than as a measure for steady state accumulation in body tissues exposed to external oxidative stress factors such as smoking or UV light exposure [120].

In the year 2000, RRS was demonstrated by Hata et al. to be a more efficient and accurate non-invasive method of measuring carotenoids in the skin [32]. After a successful proof of concept, the research sparked a wave of exploration of the various benefits of this technique in the clinic, even though it was initially thought of as a way to help prevent cancer [32]. Applications of RRS have now been extended to nutritional science, including supplement development and medical research, in which the advantages of carotenoids are constantly being investigated [30,126,127,128]. There are currently more than 10,000 portable instruments in use to monitor the efficacy of carotenoid-containing nutritional supplement formulations [127]. Carotenoid levels in the skin are rather low, requiring the laser power employed to be high, and a long acquisition time is required. The background interference can also be very high, requiring efficiently suited baseline correction algorithms to obtain the spectra [32].

Raman spectroscopy is commonly carried out on skin sites with minimal melanin and blood contents, such as the heel of the palm or foot and the tip of the finger [127]. The palm is usually preferred for carotenoid measurements because the thickness of the stratum corneum of the palm is high (~400 um) compared to other anatomical sites [32]. Carotenoid concentrations in the palm are also among the highest of all sites because of their lipophilic nature and the high lipid-to-protein ratio of the skin of the palm [32]. An additional advantage is the minimal differences in pigmentation in the palm among skin types which allows for more consistent measurements [129].

As shown in Figure 5, the skin RRS instrumentation set-up is comprised of an argon laser, a spectrograph and a light delivery and collection module [127]. A typical measurement involves the placement of the palm of the hand against the window of the module, whence it is exposed for approximately 20 s to safe laser intensities (0.16 W/cm^2^) [120]. The source laser light is delivered by an optical fibre in parallel with a lens and filtered using a narrow-band dielectric filter. It is then positioned onto the skin, where it illuminates a 2-mm-diameter spot. The resulting Raman scattered light is collected in an off-180° backscattering geometry with the aid of a second lens, which passes through a holographic notch filter and is directed to the spectrograph (connected to a computer) through a fibre bundle [120]. The resultant Raman spectrum of characteristic carotenoid peaks is superimposed on a spectrally broad and strong skin fluorescence/scattering background that the laser source simultaneously generates. The spectrum also contains weak Raman signals produced by skin components such as collagen, lipids, elastin, etc. [127,130]. The peak Raman intensity scales linearly with the carotenoid concentration at their physiological skin levels [33].

This initial instrumentation set-up, which basically used one wavelength and a small exposure diameter of 1–2 mm, has been modified over the years. For instance, a two-wavelength excitation scheme was developed some years later to include a multi-laser line system which operates both the 488 and 514.5 nm lines and special transmission filters that can enable the illumination at the two wavelengths [105,132]. The illumination area was also increased to accommodate up to 6.5 mm diameter and further eliminate the influence of inhomogeneity and pigmentation of the skin [105]. Since the efficiency of Raman scattering depends predominantly on the source wavelength and carotenoids absorption, illuminating beta carotene and lycopene in the blue and green spectral ranges, respectively, enables differentiation of the carotenoids [105]. For instance, to determine the concentrations of beta carotene and lycopene, these studies [33,132] used a multiline source and detected the Raman lines lying at 527.2 nm and 558.3 nm, each wavelength corresponding to the Raman shift of the *v*_1_ carotenoid vibration, originating from radiation at wavelengths of 488 and 514.5 nm, respectively.

One or two wavelength source Raman devices are currently commercially available that provide acceptable measurement stability but apply different measurement times and illumination spot sizes [105]. In addition to the RRS approach, some groups have used non-resonant and near-resonant approaches to analyse skin carotenoids. Lademan et al. [133] carried out in vivo Raman microscopy of cutaneous carotenoids, including topically applied carotenoids, using non-resonant conditions in the near infra-red spectral range. The source wavelength was 785 nm, which enabled an increased penetration depth compared to visible light. The concentration of the carotenoids was determined based on the *v*_1_ peak at 1523 cm^−1^. The near resonance wavelength of 532 nm has also been applied to detect skin carotenoids [134,135,136]. In the instrumental set-up described by Caspers et al. [134,135], a combination of confocal Raman spectroscopy and microscopy was employed. The laser was focused on the skin using a microscope objective. The scattered light was collected and reflected by the mirrors and a short-pass filter before being filtered by a laser-rejecting filter and focused on an optical fibre connected to a spectrometer [105,135]. This method enables the measurement of dermal carotenoids in different anatomical locations up to a depth of 30 um in the epidermis because of its high sensitivity, allowing the determination of the axial distribution of skin carotenoids at a higher accuracy (>5 um) [105]. It is, however, limited in measuring skin carotenoids because of the high absorption and scattering experienced, and the penetration depth is greatly diminished at resonant wavelengths [105,135]. The absorption and optical scattering coefficients of the skin layer can also be altered by the random distribution of chromophores in the skin layer [105]. The potential for reabsorption of light by the chromophores can also pose a challenge in this spectral range, affecting the results produced [105].

The near-infrared non-resonant range can also be challenging. Even though a reduction of absorption and scattering is possible, the Raman scattering efficiency of the carotenoids is substantially reduced because they are not resonantly enhanced in this range, making it difficult to measure very low concentrations of skin carotenoids [105].

Other variants of spectroscopy for measuring skin carotenoids have more recently emerged, such as reflection spectroscopy, which uses only a low-power white light source [126,127]. Nevertheless, the RRS approach remains the most successful and preferred method of quantifying the main carotenoids, beta carotene and lycopene in the skin, because of its non-invasiveness, high sensitivity, accuracy, reproducibility and ease of performance.

#### 5.2.2. Clinical Research Applications

Raman spectroscopy has successfully been used to selectively quantify lycopene and beta carotene in the skin using blue (488 nm) and green (514.5 nm) laser lines as sources [129]. Ermakov and colleagues were among the first researchers who used the advantage of the differing Raman cross-sectional profiles for both compounds combined with the source wavelengths to successfully quantify the carotenoids in 70 subjects [129]. Beta carotene and lycopene have different absorption values at 488 and 514 nm, which produces different scattering efficiencies at these resonantly enhanced source wavelengths [129]. These differences can be used to determine the concentration of the compounds in human skin [118]. Ermakov and colleagues further discovered that there was a wide inter-individual variation in the beta carotene to lycopene concentration ratio, which showed either of the two as the predominant carotenoid in a particular subject [129]. Findings like this can show differences in dietary patterns for different subjects and influence dietary supplementation guidance/prescription in clinics. It can also raise questions as to what factors could influence the varying ratios in the carotenoids, for instance, the different abilities of individuals to uptake the individual carotenoids or other external factors that could influence uptake. A similar study by Darvin et al. explored the concentration of the two carotenoids in the palm and other anatomical regions, such as the forearm, flexor, forehead and back of 28 volunteers. The distribution of the carotenoids was found to be strongly dependent on the anatomic site and other factors explored, such as smoking and diet [132]. There is a widespread research body using skin RRS in various areas of high clinical relevance. For instance, RRS of the skin has been demonstrated extensively to protect the skin from the production of free radicals caused by exposure to IR and UV irradiation because the skin carotenoids are strongly correlated with the magnitude of destruction of antioxidants in the skin [118,137,138]. Other areas of clinical application include diet and supplementation, disease and other stress factors, some of which are discussed in the following sections.

(a)Effect of diet and dietary supplementation.

Fruits and vegetables are a rich source of dietary carotenoids, and hence there is a growing need for suitable biomarkers of fruit and veg intake by clinicians and nutrition professionals, especially in children, for proper nutrition surveillance and interventions [30,31]. Several studies have also indicated an inverse relationship between the intake of fruit and vegetables and some life-threatening diseases, including cardiovascular diseases [139,140], type 2 diabetes [12], some cancers [6] and all-cause mortality [141]. Skin carotenoids, especially beta carotene and lycopene, are hence continuously employed as biomarkers to monitor fruit and vegetable consumption and, by implication, the general nutritional health of people [30,31]. Raman spectroscopic measurement of skin carotenoids has been shown to be efficient and reliable in monitoring fruit and vegetable intake [31]. Results from several validation studies which measured skin carotenoid levels of individuals and correlated them with dietary lifestyle reports obtained from self-administered dietary assessments and blood plasma carotenoid measurements have strongly indicated that Raman is a reliable tool for assessing the dietary status (especially fruit and veg) of individuals [30,31,123,128,142,143,144,145,146]. For most of these studies, a typical portable hand-held Raman spectroscopic device with just one laser line (usually 488 nm) was used to carry out the measurements [31,123,142,143,144,145,146]. When one study was carried out on 381 preschool children to assess their dermal carotenoid status, the parent-reported fruit and veg status positively correlated with skin RRS status (*p* = 0.02) [146]. Similarly, another study carried out on 45 children aged 8–17 established a positive correlation of skin carotenoids, as determined by RRS, with serum carotenoids (R^2^~0.5) and with reported food intake (R^2^ = 0.32) [144]. A further study of 128 children also found skin carotenoids to be correlated with both serum (0.62) and reported food intake (R = 0.4) [145].

Skin RRS has also been shown to be a reliable tool for monitoring dietary interventions. A controlled feeding intervention study carried out on 29 participants who consumed both low, high and normal carotenoid diets at different intervals for a total period of 28 weeks revealed a positive correlation between skin and plasma carotenoids at baseline (R = 0.61). The changes in diet were also reflected by skin RRS, again, in correlation with plasma levels [123].

Raman spectroscopy has been used to study the effectiveness of orally administered carotenoids/antioxidant supplements. In one double-blind, placebo-controlled randomised study, 25 volunteers on a lycopene-deprived diet received lycopene supplements for 2 weeks, after which their skin carotenoid levels were monitored using Raman spectroscopy for 2 weeks [147]. The results showed increases in cutaneous levels of both lycopene and beta carotene. There have also been reports of the high sensitivity of the method to lutein and zeaxanthin [125] and to a combination of carotenoids with other antioxidants such as vitamins E and C [124].

Since the initial proof-of-concept developments, the technique of RRS analysis of the skin has been widely accepted in the Nutritional Supplement Industry [126]. It is currently used to identify people with low skin carotenoid levels and to monitor uptake levels of individuals consuming carotenoid-rich diets and carotenoid-containing dietary supplements, thereby proving the effectiveness of both diet and supplements [126].

The accurate prediction of nutritional status has far-reaching advantages in guiding human health interventions. For this reason, advancements made in skin RRS measurements can positively influence diet-related diseases, children’s health and development and macular pigment health. Highly specific differentiation and quantification of the skin carotenoids can improve supplementation and dietary intervention strategies.

(b)Effect of stress factors

Oxidative stress commonly leads to the production of high amounts of free radicals, which are immediately neutralised by the human body’s antioxidative system and the resultant degradation of carotenoid antioxidants in the body. Many common oxidative stress factors such as illness, fatigue, smoking and alcohol consumption can be monitored using Raman spectroscopy of skin carotenoids [30,148]. Smoking, for instance, has been convincingly shown to lower dietary carotenoids [148,149,150], most likely because of the direct effects of tobacco-induced oxidation [30]. In one study by Davin et al., variation in the level of the carotenoid antioxidant beta carotene and lycopene in the human skin of 10 healthy volunteers was measured with RRS in an in vivo experiment over the course of 12 months [142]. Information on the lifestyle of the volunteers, including dietary supplementation and stress factors, was collected daily through the completion of questionnaires. The findings showed correlations between the level of carotenoids in the skin and the influence of different stress factors. Stress factors such as fatigue, illness, smoking and alcohol consumption resulted in a decrease in carotenoid levels in the skin [142]. Change in season has also been shown to affect dietary skin carotenoids [142]. In the same publication, Darvin et al. also showed that carotenoid levels in the skin were higher during the summer months and lower in the autumn/winter months for all volunteers [142].

Another factor that may be somewhat correlated with skin carotenoids is body adiposity [30]. Some studies have observed that obese subjects have lower skin carotenoids as compared to normal/overweight subjects [128,148,151], although the association in each case was not significant. Research has shown that people with high oxidative stress have lower skin carotenoid levels than those with low oxidative stress [30,120]. This finding has been correlated with skin RRS [152], suggesting that RRS scores might be used as a marker for general antioxidant status.

There are also factors such as age and sex, which may not be significant determinants of RRS skin carotenoid status in adults, although children who are older appear to have higher skin carotenoids than younger ones and babies [146]. Women also tend to have higher carotenoid status than men [128].

With regard to the influence of melanin on the RRS method, it is expected that melanin will attenuate the blue excitation as well as the backscattered Raman-shifted light, and this could potentially lead to a misinterpretation of RRS-derived skin carotenoid scores in people with higher levels of melanin [127]. However, studies have not been widely carried out in varied skin colours/races to correctly assess its dependence. Therefore, there is very limited information available on the effect of skin pigmentation on RRS measurements. One study which assessed skin colour on the inner arm by matching it to a colour wheel of skin tone samples used for prosthetic devices found that RRS measurements were lowest in subjects with the darkest skin tone. This, nevertheless, could not be correlated with other factors that vary for race/ethnicity and the multivariate analyses were not significantly associated with skin carotenoid status [128]. In general, choosing measurement sites that have a minimal influence of melanin, such as the palm of the hand or the heel of the foot, where melanin levels are low irrespective of race/ethnicity, reduces the influence of melanin on RRS measurements [30].

Research has also linked the polymorphic gene responsible for metabolising provitamin A carotenoids (like beta carotene) into retinal, beta carotene 15 15′–monooxygenase 1 (BCMO1) to beta carotene concentration in plasma [153,154]. Therefore, it can be said that genetic variation might be a significant determinant of individual carotenoid concentrations in blood.

### 5.3. Eye Health

Two major carotenoids, lutein and zeaxanthin, found predominantly in the macula of the eye, are responsible for optimising visual function and protecting the eye against light-induced oxidative damage and associated maculopathies such as AMD [21,155]. These two carotenoids, together with meso–zeaxanthin, a stereoisomer of zeaxanthin, form the macula lutea or macular pigment, a yellowish spot in the retina of the human eye [155]. Currently, there is no universal standard for measuring MP status in clinical practice; therefore, screening for the risk of ocular or neurodegenerative disease related to MP level is not routinely performed [156,157]. To accurately quantify the level of the carotenoids in the macula, gold standard methods, such as heterochromatic flicker photometry (HFP), have shown huge limitations, especially for carrying out investigations in the elderly, since it is a psychophysical test, requiring a lot of mental participation and visual acuity from a subject [158]. It can also show a high level of intra-subject variability in cases with low macular pigment densities or the presence of significant macular pathology [120]. This method measures the macular pigment optical density (MPOD) of the retina by viewing a small circular stimulus which switches between a test wavelength that is absorbed by the MP in the blue region (around 460 nm) and a reference wavelength in the green region (around 540 nm) that is not absorbed. The subject observes this change as a flicker which is then reduced to a zero reference point by adjusting the intensity of the test wavelength while viewing the stimulus centrally and then peripherally. Other methods based on reflectance or autofluorescence have also shown great potential for measuring the macular pigment, but they are expensive and widely inaccessible [120,158].

#### 5.3.1. Advantages of Raman Spectroscopy for Macular Pigment Detection

In Raman spectroscopy, the strong absorption bands of carotenoids greatly enhance the specific Raman response for increased sensitivity of detection [159]. The method was first reported by Bernstein et al. in 1998 when the group measured carotenoid levels in flat-mounted human retinas and eye cups in experimental animal eyes. Raman spectroscopy was found to be rapid, repeatable, highly sensitive and specific. The signal strength recorded also correlated with macular carotenoid concentration as measured by HPLC [160]. Shortly after, validations were carried out in living human retinas in a study involving a large number of elderly subjects with and without AMD [161]. Because of its several advantages, there is now a growing body of research that advocates the use of Raman spectroscopy in the detection of macular pigment carotenoids for the early diagnosis of eye diseases such as AMD, glaucoma and associated maculopathies [162].

Carotenoids have particularly intense absorption bands in the blue/green wavelength range [163]. In the ocular tissue, carotenoids show the strongest resonant enhancement using a blue laser as a source (488 nm) [163]. Therefore, when used in vivo, the spectra are free from conflicting signals as no other ocular biological molecules present in significant concentration show similar resonant enhancement at this wavelength [158,163].

In principle, the positioning of the macula carotenoids, concentrated in the Henle fibre layer, a very thin membrane (thickness of only 100 um), provides a chromophore distribution resembling an optical thin film which makes it easy to minimise self-absorption of illuminating or scattered light [120,161]. Another advantage of Raman analysis of the macular is the good clarity from the ocular media (cornea, lens and vitreous) in the spectral range of the source and scattered radiation (488–527 nm). The media clarity is enough and should generally not affect the Raman signal, requiring appropriate correction factors only in extreme visual pathologies such as the presence of cataracts [120,161]. The third advantage of Raman spectroscopy is the positioning of the macular carotenoids which are located anteriorly in the optical pathway through the retina, causing the illuminating and backscattered light to never encounter any highly absorptive pigment such as photoreceptor rhodopsin and retinal pigment epithelium (RPE) melanin [161].

When compared with other non-invasive methods such as heterochromatic flicker photometry (HFP), Fundus reflectometry and auto fluorescence, because of its very high sensitivity, RRS can detect an increase in foveal carotenoids which could be easily missed by these gold standard methods [158]. RRS has been shown to be rapid, repeatable (repeatability range of ±10%) and well-accepted by patients in measuring macular pigment carotenoids [161,164]. It provides an absolute measurement of the macula carotenoids without the need to have a peripheral zero reference point, as the traditional methods require [161,164]. RRS and HFP have also been compared in different studies, and very significant correlations have been reached [158,165].

The limitations of RRS in macular detection at first was the fact that the detector measured a single spot of 1 mm area, which was very small, compared to measuring a wider spatial area which could hold even more information and hence improve accuracy. This initial instrumentation has since been modified to include a spatial imaging system [166]. The resonance Raman imaging system was first described by Gellermnan et al. [166] as a higher-resolution optical imaging technique that can quantify macular pigment carotenoids as well as display spatial variations in MP distribution [91,158,167].

Figure 6 shows the Raman intensity maps of the macular carotenoids lutein and zeaxanthin as well as the MP total carotenoids, obtained using the resonance Raman imaging system. The instrumentation employs an argon laser at 488 nm, a charge-coupled device (CCD) camera array in combination with a narrow bandpass angle turnable filter and a well-suited light delivery and collection module [166]. It can produce a topographic two- and three-dimensional pseudo-colour map of macular pigment distribution at less than 50 μm resolution. It does this by subtracting two images, the first one taken with the filter aligned to transmit the 1525 cm^−1^ carotenoid peak and the second with the filter rotated a little to transmit light a few wavenumbers away. A portion of the excitation light is split by a beam splitter to a second CCD camera to capture an image of the source laser intensity distribution [120,129,167,168]. Further advantages of this method include the ability to measure a wider retinal spot size of up to 3.5 mm in diameter, compared to the initial limitation of 1 mm [167]. Imaging can also be carried out in both dilated and non-dilated pupils. The method does not depend on the reflection of light at the sclera. Any overlapping background signals from the lens and deeper layers can be easily subtracted from the overall light response. Additionally, it does not require a reference point, and no assumptions need to be made apart from approximating the spectrally broad background with the background response at a wavelength that is slightly offset from the MP Raman response. This means the measurement recorded is the absolute MP concentration distribution in the macular region [91,166,167,168]. These advantages can potentially aid investigations into correlations of MP patterns with developing pathology. It can also track MP level and spatial changes in human subjects related to dietary modification or supplementation [166,167,168].

#### 5.3.2. Clinical Research Application

Initial explorative studies of Raman spectroscopy of the retina involved taking spectra from specimen eye tissue, including flat-mounted human cadaver retinas, human eye cups and frog eyes, using a liquid nitrogen-cooled lab grade high-resolution Raman spectrometer coupled with a high-powered argon laser [160]. This initial setup was unsuitable for use in living humans. The instrument was modified for high light throughput and a lower spectral resolution. The device had a low-powered air-cooled argon laser which projected a 1 mm 0.5 mw 488 nm spot onto the foveal region for half a second through a pharmacologically dilated pupil [161,169]. Subjects were asked to fixate on a target while measurements were recorded. The instrument was calibrated using a solution of lutein and zeaxanthin. The detector response could measure the optical density of up to ~0.5, after which saturation began to occur due to self-absorption affecting the accuracy of the reading. To make up for any of these errors, a calibration curve was used at higher physiological levels of the macular pigment parameters [120,129]. Carotenoid measurements obtained showed that older people had lower concentrations. This was observed in subjects who were not supplemented with lutein or zeaxanthin, and the results were consistent with those obtained from known MP measurement techniques [120,129]. Declines observed in carotenoid levels of individuals can help predict the onsets of macular pigment diseases such as AMD [161]. The ability to accurately measure macular carotenoid levels has helped to affirm the significance of consuming carotenoid supplements for patients with macula myopathies [170]. Dietary supplementation with foods or supplements abundant in lutein or zeaxanthin has been shown to increase macular pigment levels and lower the risks of AMD [161,171]. For instance, patients with AMD who consumed at least 4 mg/day of lutein supplements over three months had improved macular pigment levels compared to those who did not take the supplement [164,172]. Lutein and zeaxanthin supplements are hence currently promoted by clinicians and commonly found over the counter at pharmacies, nutritional stores and optometry practices.

RRS has also been applied to study the depletion and progression of carotenoids related to other macular diseases such as Stargardt disease and macular drusen [173]. Zhoa and colleagues carried out a series of macular resonance Raman measurements on patients with different macula dystrophies, including Stargardt disease [173]. Macular levels were measured using an argon laser operating at 488 nm. Carotenoid peak intensities at 1525 and 1159 cm^−1^ were reduced in patients with Stargardt compared with normal patient’s eyes [173]. A similar study was carried out on patients with macular drusen and also indicated reduced macular carotenoid levels [164].

Despite these advances, clinicians and researchers do not fully understand how the resonance Raman technology can be translated effectively into the clinical setting for monitoring and tracking the concentration of macular xanthophyll pigments in patients at risk of macular disease. Much of the studies conducted have been based on quantifying the carotenoids contained in the macular pigment as a whole. Differentiating the individual carotenoids and quantifying them can bring further insight into understanding the extent of the macular carotenoid distribution and their specific importance in controlling macula diseases. While this is possible using HPLC in extracted tissue, it becomes more difficult with non-invasive techniques such as Raman spectroscopy and more so in living human eyes because of the high similarity of lutein and zeaxanthin. Recently, Li et al. used a high-resolution confocal Raman microscopic mapping in combination with multivariate algorithms to measure the individual distribution of lutein and zeaxanthin in human retinal sections and flat-mounted human retina [91]. Their results showed that zeaxanthin was highly concentrated in the fovea, extending from the inner to the outer limiting membranes, with especially high concentrations in the outer plexiform layer, while, on the other hand, lutein was much more dispersed at relatively lower concentrations, suggesting that zeaxanthin may be of more significance in human macular health and disease [91].

### 5.4. Blood

#### 5.4.1. Whole Blood and Blood Cells

Blood serum/plasma is the most widely studied bodily fluid and primary specimen of interest in clinical diagnostics [174,175]. It contains carotenoids in small quantities (>3 μmol/L) [17,176], amongst many other biological components of analytical importance. Raman spectroscopy can detect carotenoids in whole blood as well as in serum/plasma, from which the cells have been separated [96,97,98,99]. For instance, a near resonance Raman approach has been used to detect the presence of carotenoids in whole blood [96]. In a study by Casella and colleagues, the three carotenoid peaks were clearly enhanced in resonantly enhanced and surface-enhanced Raman spectroscopic analysis using a 514 nm near resonance laser source [96]. The addition of colloidal silver nanoparticles significantly increased the Raman signal, providing a clear SERS spectrum of whole blood [96].

Raman spectroscopy has also been used to detect carotenoid levels in lymphocytes from peripheral blood [177]. The study of Ramanauskaite et al. used microspectroscopy to measure carotenoid levels (specifically, beta carotene) in gall bodies of human lymphocytes from healthy subjects and lung cancer patients [177]. A 660 nm laser was used as a source, and an accumulation of carotenoids in the gall bodies of the lymphocytes was revealed [177]. The results from this study show carotenoids are present in high concentrations in the cytoplasm of specific cell populations. This could shed more light on the specific role of carotenoids in the immune system, which may lie in the actual nature of their function or in their location. White blood cells play an essential role in the immune system [178], and high levels of carotenoids in these cells may imply they help to execute antioxidant and protective properties in the body. Quantifying the carotenoids found in these cells can, for instance, provide supporting information on the antioxidant status of an individual and perhaps influence clinical decisions.

Another study by Burkur et al. explored the effect of IR light on various blood cells. Whole blood cells (lymphocytes and platelets) were analysed using the near IR wavelength of 785 nm [101]. Raman tweezers were used to measure and assign micro-Raman spectra of optically trapped, live red blood cells (RBCs), white blood cells (WBCs) and platelets. The results revealed that both haemoglobin and the cell membrane sustained damage. Amongst all the components of the blood, lymphocytes and platelets had the most drastic change in the peaks corresponding to beta carotene (identified at 1156 and 1515 cm^−1^), which the authors suggested might most likely be due to the possibility of free radical-induced damage of beta carotene in lymphocytes and platelets [101].

#### 5.4.2. Clinical Research Application

Raman spectroscopy of carotenoids in the serum has also been explored in respiratory diseases, carcinomas and parasitic and organ diseases [97,98,99,100,179,180,181,182,183,184,185]. In many of these clinical studies, carotenoid signals were assigned to beta carotene, although they could equally be assigned to other prominent serum carotenoids such as lycopene, lutein and zeaxanthin, which have almost identical Raman features. However, the primary interest of some of these diagnostic studies was to identify obvious changes (if any) in carotenoid levels as a result of the disease in question and so differentiating and perhaps quantifying the carotenoids was not of primary importance. For instance, in the detection of respiratory diseases such as asthma and tuberculosis in serum samples, Raman spectroscopy has been used to show the involvement of carotenoids. In the case of asthma, excitation was enhanced resonantly at 532 nm, and the carotenoid peaks observed in healthy patients’ serum were clearly diminished or suppressed in asthmatic patients’ serum [97]. In a Raman analysis of tuberculosis patient serum, multivariate analysis was combined with Raman excitation at 532 nm [98] and 785 nm [99]. In both cases, there was an observably prominent decrease in the intensity of all three characteristic carotenoid peaks in affected patient serum compared to cured and healthy patients.

There is a large body of evidence supporting the consumption of carotenoids to help reduce the risk of some cancers, including breast, oesophageal and lung [4,5,6,7]. This is mainly due to their unique antioxidant characteristics. While cancer remains a leading cause of fatality, researchers and clinicians continue to explore efficient ways of tackling this disease. Serum carotenoids were reduced significantly in the cancers implicated. In patients with nasopharyngeal cancer, the Raman analysis of serum combined with multivariate techniques showed an obvious decrease in the intensity of the 1519 cm^−1^ carotenoid peak, observed at 785 nm excitation in the serum of positive subjects [100]. Another study showed a decrease in carotenoid peaks observed from the Raman analysis of serum from patients with oral cancer. Here, the spectra obtained were resonantly enhanced by the 532 nm laser employed, showing notable variance in the 1155 cm^−1^ and 1523 cm^−1^ peaks [186]. Another study exploited Raman spectroscopy to characterise the biomolecules present in the blood plasma of clinically confirmed normal, premalignant (oral sub mucous fibrosis) and malignant (oral squamous cell carcinoma) conditions using a 784.15 nm laser line. Raman spectral signatures revealed relatively less intense carotenoid peaks for the malignant group than those of the normal group [187].

In addition to oral cancers, oesophageal cancer has also been detected using Raman analysis of blood plasma. This was conducted using a specially designed quartz capillary tube as a sample holder to obtain high-quality resonance Raman spectra, which revealed reduced carotenoid peak intensities in positive patients [180].

In patients with colorectal cancer, near-infrared Raman spectroscopy (785 nm) detected significantly lower intensities of carotenoid peaks (1155 cm^−1^ and 1523 cm^−1^) in dried serum samples of positive subjects [182]. A resonantly enhanced Raman approach has also been carried out in patients with colorectal/rectum cancer, and similar results were obtained [182]. Here, a source wavelength of 488 nm was applied to serum samples of both normal and cancer patients. The Raman measurements, in combination with multivariate analysis, revealed significantly reduced carotenoid peaks (1170 cm^−1^ and 1538 cm^−1^) in cancer patients compared to normal patients [182]. This suggests significant potential for the use of Raman spectroscopy of serum in colorectal cancer diagnosis.

In leukaemia research, Raman spectroscopy has been applied to distinguish between normal and leukaemia blood serum and to identify the different types of leukaemia based on serum biochemistry. The research carried out by Gonzalez-Solis et al. obtained blood samples from seven clinically diagnosed patients with three leukaemia types and analysed them using an 830 nm laser source [183]. The result revealed the serum samples from patients with leukaemia and from the control group could be discriminated against with the help of multivariate statistical methods of principal component analysis (PCA) and linear discriminant analysis (LDA). When compared to control samples, PCA and LDA clearly differentiated the samples and further differentiated the three leukaemia types. This successful discrimination was largely due to the presence of two prominent carotenoid peaks at 1523 and 1160 cm^−1^, which were less intense in leukaemia serum spectra, amongst other biochemical factors [182]. This suggests again that Raman spectroscopy along with multivariate analysis, can become a very relevant technique in detecting and identifying the different leukaemia types using serum samples. In all the cancer studies explored, there seems to be great potential for the use of Raman spectroscopy of carotenoids in cancer detection. The correct identification and quantification of serum carotenoids implicated in a cancer type can perhaps influence early diagnosis. It can further guide the application of preventive measures such as precise dietary supplementation recommendations.

Raman spectroscopy of the serum has had an even further positive impact on other less common diseases. An example is a recent report using Raman analysis of carotenoids in serum to detect parasitic diseases. A study in 2021 described the use of resonance Raman in combination with PCA-LDA to detect echinococcosis, a parasitic disease caused by infection with tapeworms [184]. The detection of two resonantly enhanced beta carotene peaks (1154 cm^−1^ and 1515 cm^−1^) with significantly decreased intensity in the diseased subjects as compared to healthy ones was the basis of discrimination [184]. A similar study used the same approach to distinguish between echinococcosis and liver cirrhosis, two different diseases that have very similar symptoms, making them challenging to distinguish [185]. Raman spectra of serum samples from echinococcosis, liver cirrhosis, and healthy volunteers were recorded using 532 nm excitation. The normalised mean Raman spectra showed specific biomolecular differences associated with the disease, mainly manifested as reduced carotenoid peaks (1154 cm^−1^ and 1515 cm^−1^) in the serum of patients with echinococcosis and liver cirrhosis, compared with those of normal subjects. The further application of multivariate techniques, such as PCA and LDA, distinguished patients with echinococcosis, liver cirrhosis and healthy volunteers with an 87.7% accuracy [186]. This work again demonstrates the potential for the non-invasive identification of echinococcosis and liver cirrhosis and the limitless potential of Raman spectroscopy of the serum/plasma in the detection of disease conditions.

## 6. Correlations between Raman Spectra of Macula, Skin and Serum Carotenoids

Ocular and systemic measurement and imaging of the macular carotenoids lutein and zeaxanthin have been employed extensively as potential biomarkers of AMD risk [161,162,171]. Imaging methods such as retinal autofluorescence imaging (AFI) and heterochromatic flicker photometry (HFP) measurement of macula pigment optical volume have been significantly correlated with skin RRS in various studies [158,165,187]. For instance, skin RRS measurements of beta carotene and lycopene correlated strongly with AFI macular pigment volume under the curve (MPVUC) measurements and HPLC-measured serum carotenoid levels (particularly, serum zeaxanthin) in a study by Conrady et al. carried out in a clinic-based population [187]. The measurements obtained were reproducible and not significantly affected by the presence of cataracts [187]. There have also been several studies on correlations between RRS skin carotenoids and HPLC serum carotenoids in areas such as nutrition [31,123,144,145]. Many groups have researched correlations between carotenoids in diet and nutrition studies both in children [144,145] and adults [31,123]. An example is a study carried out by Jahn et al., which found a significant correlation between skin carotenoids as measured by RRS and serum carotenoids measured by HPLC in adults investigated for fruit and veg intake [31]. In terms of supplementation, the study carried out by Blume-Peytavi and colleagues, in which subjects involved were placed under a lycopene-deprived diet and offered a lycopene supplement instead, showed that both skin RRS and serum HPLC were highly responsive to supplementation, causing an increase in both lycopene and beta carotene as well as lutein and zeaxanthin [147]. RRS measurements taken from the forehead showed the closest correlation to lycopene variation in plasma [147]. Lycopene skin levels were found to be less sensitive to supplementation compared to plasma. Beta carotene plasma levels were also not significantly influenced by lycopene intake. Skin levels of beta carotene were stable under lycopene deprivation and increased with supplementation [147].

In general, efforts to correlate carotenoids measured by RRS in the skin and by HPLC of serum seem to have been the most common and as such, findings have been used to help establish the antioxidant status of patients [30,120]. Skin carotenoids have also significantly been correlated with serum lycopene, alpha carotene and beta carotene in control and subjects not positive for cardiovascular diseases, which are characterised by low antioxidant levels [151]. There have also been studies in cattle in which a significant correlation between skin RRS and serum was also established [46]. However, to date, there have been no significant correlations between the RRS of the carotenoids of either skin or blood with those of the retina.

## 7. Limitations and Future Perspectives

The Raman spectroscopic method of measuring dietary carotenoids comes with numerous advantages. Its emergence into the disciplines of nutrition and medicine has brought immense progress to various aspects of both fields. The method is safe and rapid; measurements can literally take seconds to perform. Its high sensitivity means that it can be used to measure very low concentrations of carotenoids, which might be challenging to carry out using other methods. Its durability means it does not necessarily require very heavy machinery, so it can be made compact for easy transportation. The Raman technique has been shown to measure carotenoids in various systems of the body, including tissue sections of organs and bones [107,110,117]. Skin carotenoid measurement is probably the most popular because of its non-invasive and diverse applications. Its utility in measuring food and nutrition status, especially in children, continues to advance as more clinicians open up to using Raman spectroscopic devices for skin carotenoid measurements in clinic settings. Raman spectroscopic analysis of carotenoids in the skin has been applied widely in the clinical setting as a rapid, objective and non-invasive antioxidant/biomarker quantification technique. There are now applications in ophthalmology for determining the macular pigment health of both newborn infants and young children. Here, macular pigment optical density is measured alongside skin RRS to determine carotenoid status [188,189]. In otolaryngology, skin carotenoids are measured non-invasively using catheter-coupled RRS, and the results are assessed for protection against cancer-causing agents [126]. In neonatology, RRS measurements of skin carotenoids can act as useful biomarkers for serum carotenoid levels, which can be problematic to obtain due to the limited blood volume in newborn infants [190]. They can also help to understand the risk for retinopathy of prematurity in newborns by serving as predictors of macular carotenoid levels in the developing retinas of these infants [189]. They can further help to understand how maternal carotenoid status can influence that of the newborn, especially as it relates to the maternal nutritional status [188]. In epidemiology studies, deviations from normal levels in skin carotenoid levels measured by RRS can indicate potential disease conditions, including metabolic conditions, nutritional health, cardiovascular dysfunctions and tumorigenesis [126].

In orthopaedic medicine, skin carotenoid status, as measured by Raman spectroscopy, can be used as a biomarker to ascertain the optimum health and homeostasis of bones. Bone formation involves a number of balanced biochemical reactions that rely heavily on food intake and nutrition. For instance, a high intake of fruits and vegetables has been shown to reduce the risk of osteoporosis [191]. Carotenoid antioxidants have also been described to be beneficial micronutrients for the maintenance of normal bone metabolism, and carotenoid and lycopene intake has been shown to have a protective effect on fractures of the hip [192]. Since it has been properly demonstrated that all carotenoids contained in the skin are found in the bone as well [107], there is limitless potential for carotenoid levels in the bone to be correlated with skin levels to establish a sound RRS carotenoid marker system for determining bone carotenoid status.

Many of the RRS devices in circulation use a single-wavelength source scheme, usually 488 nm [31,123,142,143,144,145,146]. At this wavelength, skin carotenoids come into resonance with very minimal overlap with other skin components. Analysing the intensity of the C=C carotenoid peak provides information on the concentration of all cutaneous carotenoids at the same time. However, while the influence of other carotenoids, such as lutein and zeaxanthin, on skin measures might seem negligible, it can significantly influence the Raman spectra when a subject has high lutein or zeaxanthin levels, which can happen, for instance, if they are consuming a dietary supplement. The reading obtained might then not be absolute for beta carotene and lycopene. There can also be the issue of reabsorption of scattered light by cutaneous lycopene, which again can reduce Raman efficiency [105]. The two-wavelength set-up, which measures beta carotene and lycopene independently, can become hugely influenced by carotenoids with similar absorption spectra and scattering profiles to beta carotene. These include lutein and zeaxanthin, together with their isomers. Other carotenoids such as alpha, gamma and sigma-carotene can also influence the Raman signals of beta carotene.

Even though there is ample evidence showing beta carotene and lycopene as the dominant skin carotenoids and these two can be distinguished with a degree of exactitude [132], the extent of influence of similar carotenoids on the Raman spectra of cutaneous carotenoids is not known. A more stringent identification modification to the technique, whereby the conflicting carotenoids can be distinguished, may improve the accuracy of the results produced in skin RRS. For instance, single wavelength devices are promoted over multiple wavelength devices for the obvious reason of reducing the influence of other carotenoids or conflicting substances. Similarly, the analysis of the intensity of a Raman peak at 527.2 nm, which provides information on all skin carotenoids, has also been reported to improve the specificity of Raman measurements carried out with single-wavelength 488 nm devices [105]. The possible reabsorption of scattered light by lycopene or beta carotene, which can reduce the intensity of scattering, remains a common issue; hence there are several multivariate techniques for subtracting the difference suited to different Raman set-ups. Another option that can be considered is to measure lycopene alone instead of in combination with beta carotene. This can be conducted in the green excitation region with an excitation source of approximately 516 nm [105,130]. The strong excitation of lycopene will also dominate over other carotenoids. At this wavelength, the possibility of reabsorption is also eliminated [105,130]. Despite the advances in skin RRS, many of these commercially available devices differ in measurement time and illumination spot size, leading to considerable variability in quantification measurements [105]. A more efficient application of the technique would therefore involve the standardisation of measurement and analysis conditions across clinics.

RRS measurement of the ocular carotenoids shows great potential in comparison to existing methods. It directly measures absolute amounts of carotenoids in the illuminated regions, providing the exact carotenoid concentration distribution in the macular region of the eye. It is also highly specific and can be established as a label-free tool for reliably quantifying macular carotenoids in patients. Macular carotenoids are usually measured together even though they consist of lutein, zeaxanthin and meso-zeaxanthin. Despite the fact that they provide protection for the eyes in their combined state in macular pigment, information on the specific influence of the individual carotenoids might play a pivotal role in enhancing MP research and eventual translation into the clinic. For instance, while electron paramagnetic resonance spectroscopy studies of carotenoid mixtures in solution indicated that a mixture of meso-zeaxanthin, zeaxanthin and lutein in a ratio of 1:1:1 provides a more efficient quenching of singlet oxygen than the individual carotenoids [193], recent Raman spectroscopic mapping results revealed that zeaxanthin is more concentrated in the region of the fovea than lutein, which is more diffusely distributed, at relatively lower concentrations, which led the authors to speculate that zeaxanthin may play a more important role in maintaining the health of the MP than lutein [91]. If, potentially, one carotenoid has a more significant influence on the macula status than the other, this can greatly advance interventions and treatment strategies. Differentiating and quantifying macular carotenoids can therefore serve as a magnifying glass to further understand and harness more specific information on the MP status of an individual. It can further guide more targeted nutrition and supplement-based strategies. The high specificity of RRS means it has the potential to monitor slight changes in the macular pigment, making it possible to detect early onsets of diseases such as AMD, which have posed a significant challenge for clinicians in terms of early diagnosis. As a potential early diagnostic tool, it can help stop the progression of macular diseases and avert potential blindness through the implementation of early treatment strategies. However, clinicians are still trying to understand how to implement reliable and quantitative methods for analysing carotenoid data from macular pigment measured by RRS in the clinic, and although the technique progressed to clinical trials in the US in 2003, the results were not published, and it is still not yet established as a routine technique [194]. The spectral quality, and therefore sensitivity and specificity of the RRS technique is dependent on the source laser power, which maybe be a challenge for in vitro retinal monitoring.

Blood carotenoid RS measurements are the most limited; the obvious issue is that blood has to be drawn from a person. This is not ideal in most environments and field-based studies because of, for instance, the need for a trained phlebotomist, a sterile environment, blood processing equipment and standard laboratory conditions, not to mention difficulties and trauma that may arise from obtaining samples from children, preterm babies or other physically compromised individuals. There is also the issue of the cost of sample collection, processing and storage, most of which can be avoided with a non-invasive technique. The need for study participants to agree to venepuncture can also be an issue, and this can slow down research and clinical trials due to inadequate samples available. Because of these shortcomings, the non-invasive method, i.e., skin RRS, is preferred over blood RRS, even though there is enormous potential in using serum/plasma for RRS carotenoid quantification. Despite this, blood still remains the primary specimen of interest in clinical diagnostics because of its high analytical importance [173]. Additionally, blood RS only requires very small amounts of the sample that can be obtained, for instance, from blood draws collected for other procedures, unlike with HPLC analysis which would require higher volumes of samples. Therefore, it is imperative to find ways to minimise the invasiveness of the blood collection procedures, perhaps exploring advancing whole blood analysis, which can incorporate only a drop of blood from areas such as the heel or thumb, such as the finger stick method used with diabetic patients. Even though a lot of HPLC serum measurements correlate with both skin and macular RRS, RRS measures of carotenoids will be even more useful if there is a significant correlation, for instance, between blood and macula carotenoids, lutein and zeaxanthin. A single blood sample could be used for carotenoid measures as well as other blood investigations that may be required, and results can be applied to assess macula health without carrying out additional RRS eye measurements. It was demonstrated recently that the dietary carotenoids lutein, zeaxanthin and beta carotene could be differentiated and quantified in serum albumin [66]. Advancements in this direction can further explore the usefulness of blood Raman spectroscopy measurements as a reliable diagnostic factor in both macular pigment status and general nutritional health.

## 8. Conclusions

Raman spectroscopy has been employed quite vastly in the analysis of carotenoids contained in different biological systems, including eyes, skin and blood, with lutein, zeaxanthin and beta carotene counting amongst the most widely researched. The high potential for employing carotenoids as biomarkers in nutrition and eye health has attracted increasing attention over the last few years. Advancements in the RS of carotenoids provide very good and reliable qualitative analysis of the carotenoids, for example, in simply differentiating normal from disease conditions, and so can be applied in the diagnosis of various diseases including cancers, heart conditions, respiratory diseases and diabetes mellitus. Despite clinical trials, the use of RRS for routine in vitro retinal screening has not come to fruition. Measurement of retinal carotenoids in the skin and/or human blood serum may therefore be a more realistic option in the short term. Future work in differentiation and quantification of the carotenoids using Raman spectroscopy could hasten the translation of such a technique into clinics

## Figures and Tables

**Figure 1 molecules-27-09017-f001:**
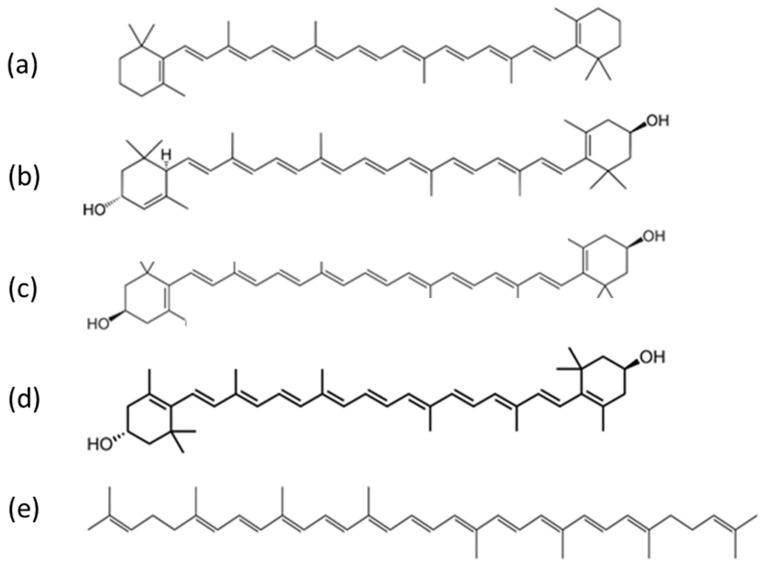
Examples of carotenoid compounds of relevance to human health: (**a**) beta carotene, (**b**) lutein, (**c**) zeaxanthin, (**d**) meso-zeaxanthin and (**e**) lycopene (public domain images reproduced from Wikimedia Commons).

**Figure 2 molecules-27-09017-f002:**
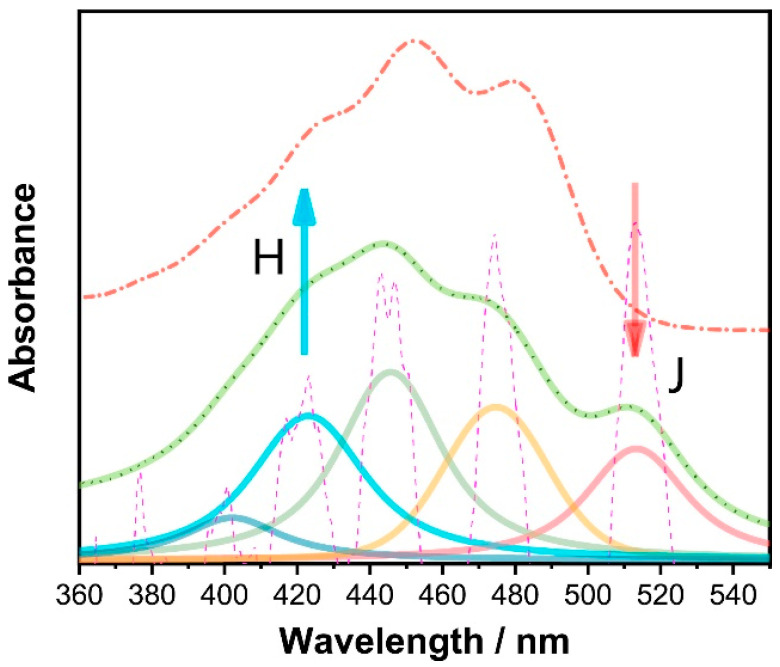
Absorption spectra of beta carotene aggregates in 1:1 aqueous ethanol (dot) and in pure ethanol (dash dot). Dash line represents the negative second derivative of the experimental results. Coloured lines are the Voigt fitting components and their sum (green) (Reprinted with permission from Ref. [54]. 2022, Elsevier).

**Figure 3 molecules-27-09017-f003:**
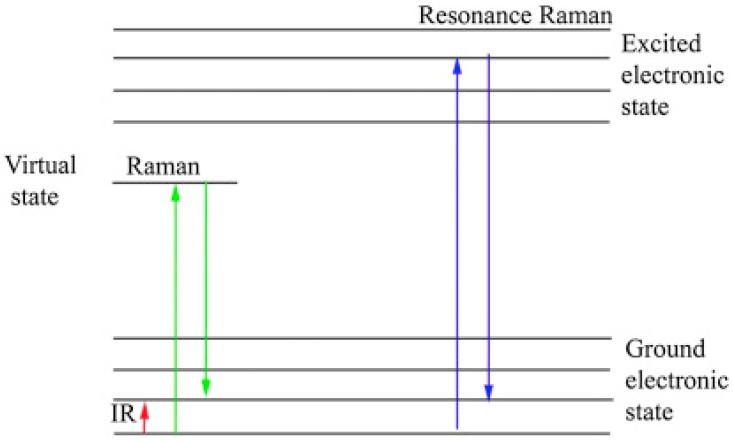
Schematic depiction of non-resonant and resonant Raman scattering processes. Energy levels are represented by horizontal lines, and the vertical arrows show the transition from one state to another (Reprinted with permission from [85]. 2022, Elsevier).

**Figure 4 molecules-27-09017-f004:**
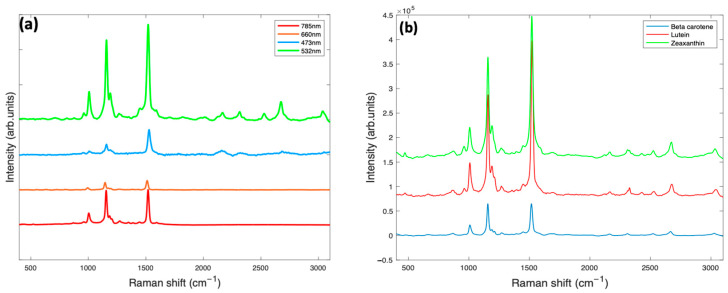
(**a**) Beta carotene spectrum at multiple wavelengths and (**b**) 532 nm spectrum of beta carotene, lutein and zeaxanthin.

**Figure 5 molecules-27-09017-f005:**
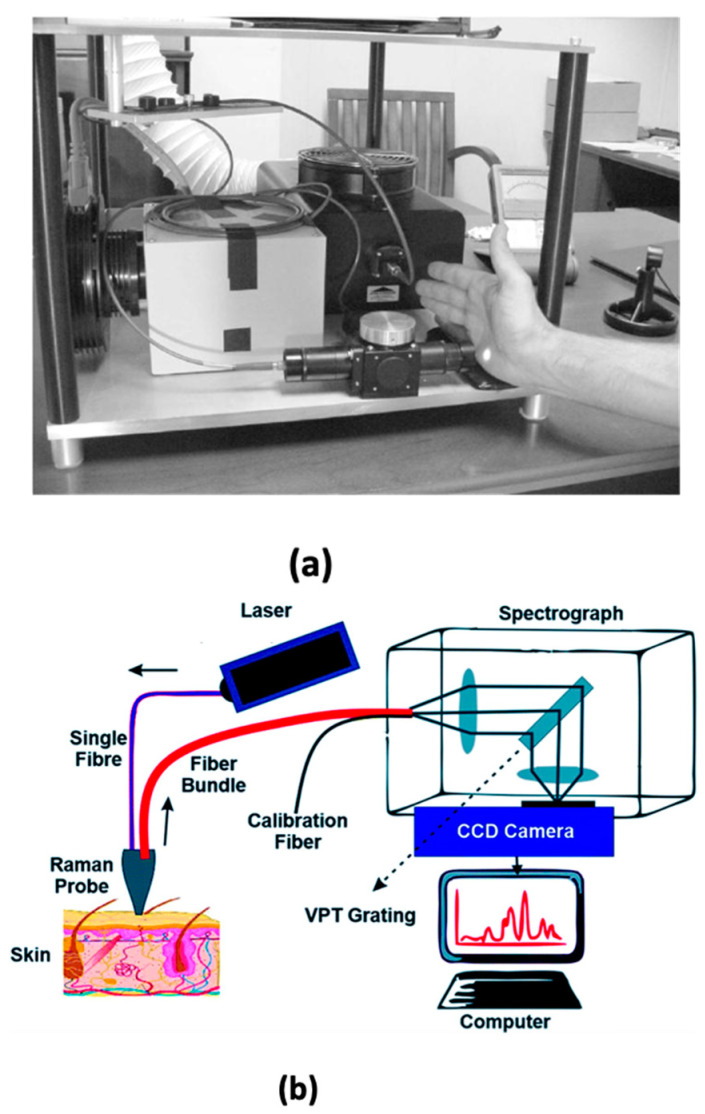
(**a**) Raman spectroscopic detector showing the argon ion laser, spectrograph, light delivery/collection module and excitation laser spot on the palm of a subject (Reprinted under the Creative Commons Licence from Ref. [119]. 2022, MDPI). (**b**) Diagrammatic description of the Raman spectroscopic detector on a skin surface [131].

**Figure 6 molecules-27-09017-f006:**
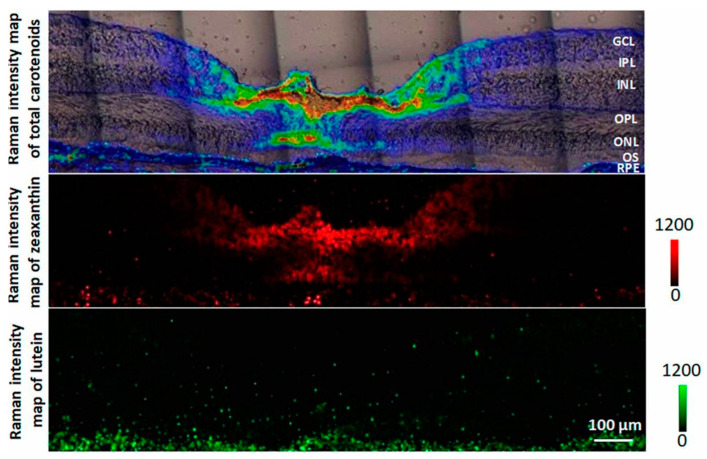
Raman intensity mapping showing the distribution of total macula carotenoids and individual macula carotenoids, zeaxanthin and lutein in a human retinal section (Reprinted with permission from Ref. [91]).

**Table 1 molecules-27-09017-t001:** Mean carotenoid concentration in human blood plasma [17].

Carotenoid	Concentration (μmol/L)	
Lycopene	0.62 ± 0.01	*n* = 56 studies
Beta Carotene	0.47 ± 0.01	*n* = 78 studies
Lutein/Zeaxanthin	0.31 ± 0.01	*n* = 31 studies
Cryptoxanthin	0.17 ± 0.01	*n* = 44 studies
Beta Carotene	0.12 ± 0.01	*n* = 53 studies

**Table 2 molecules-27-09017-t002:** Carotenoid composition of human skin in ng/g tissue (Reprinted with permission from Ref [33]. 2022, SPIE).

Carotenoid	Skin Source			
	1	2	3	Mean
Lycopene and Z-isomers	105	9	93	69
Carotenes (α, β, γ, ξ)	96	8	55	53
Lutein and Zeaxanthin	26	ND	ND	9
Phytoene and Phytofluene	113	51	74	79
Total	340	68	222	210

ND = Nondetectable.

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
