# Peer review of "Raman Spectroscopy of Carotenoid Compounds for Clinical Applications—A Review"

_molecules, 2022, doi:10.3390/molecules27249017_

Round 1

Reviewer 1 Report

Report on Manuscript molecules- 2072319. Title: Raman Spectroscopy of Carotenoid Compounds for Clinical Applications – A Review. Authors: Joy Udensi, James Loughman, Ekaterina Loskutova, Hugh J. Byrne Submitted to section: Analytical Chemistry.

I always welcome these type of Review articles, whenever I’m interested in a project in the subject in revision. They are always very useful for the interested scientists working in the use of the same field or when planning to adapt a given optical or spectroscopic methodology successfully applied to biomedical themes. As a good example of this type of Review Articles, the present manuscript provides very useful information for all applied scientists who will be interested in clinical applications of the quantification of the most important carotenoid compounds by Raman spectroscopy (RS), specially lycopene, lutein and zeaxanthin (and mainly these last two) in either blood serum or in the palm skin of subjects scrutinized for some clinical aim.

I find the article extremely well written and covers very broadly many clinical applications of RS: I specially find extremely interesting an very well covered an described the applications to quantify the amounts of lutein and zeaxanthin in the macular pigment around the macula in the retina, as well as their relevance in nutritional effects as measured by the quantity of carotenoids, as a result of the consumption of fruits in the skin of the subjects studied. I equally find very interesting and promising the application to correlate the diminution of carotenoids concentrations in blood serum, for several types of cancers.

Hence, my conclusion is that the present is a very pertinent review article and should be published.

My only observation that I think may help to improve the article for the scholar or clinical professional who reads the article to get immersed in some research related to the scope of this article is the following:

I strongly advice that a Table is added where some of the relevant physical or chemical parameters of the 5 carotenoids compounds subject of the manuscript. For instance, parameters such as the conjugation length, mentioned in lines 118 and 126, should be included, as well as summarizing their relevant absorption maxima, Raman lines, discussed in section 2.2, or any other parameter relevant for the researcher to consider when probing these compounds.   

Author Response

Please see the attachment. Responses are written in red font

Reviewer 2 Report

The authors provide a comprehensive review of carotenoid compounds for clinical applications. The overview is exhaustive ranging from structural and optical properties of carotenoid, via an introduction to Raman spectroscopy to applications in nutritional analysis and medicine. This review is well written and the topic is highly interesting. It can almost be accepted as is, but I have a few minor remarks:

1.) lines 172/173: The first sentence about Raman spectroscopy is quite vague. In which sense is energy coupled to vibrations?

2.) line 179: It is correct that the energy difference observed is equivalent to that in IR, but I would expect a statement about the complementarity of the two methods then.

3.) The titles of sections 4 and 5 are the same, this should be corrected.

4.) Starting from page 20, there are still a lot of mark ups from editorial changes. This should be corrected.

Author Response

Please see the attachment. Responses are written in green font
